# Internal or External Training Load Metrics: Which Is Best for Tracking Autonomic Nervous System Recovery and Function in Collegiate American Football?

**DOI:** 10.3390/jfmk9010005

**Published:** 2023-12-21

**Authors:** Eric Renaghan, Harrison L. Wittels, S. Howard Wittels, Michael Joseph Wishon, Dustin Hecocks, Eva D. Wittels, Stephanie Hendricks, Joe Girardi, Stephen J. Lee, Samantha M. McDonald, Luis A. Feigenbaum

**Affiliations:** 1Department of Athletics, Sport Sciences, University of Miami, Miami, FL 33136, USA; eric.renaghan@miami.edu (E.R.); lfeigenbaum@med.miami.edu (L.A.F.); 2Tiger Tech Solutions, Inc., Miami, FL 33140, USA; hl@tigertech.solutions (H.L.W.); shwittels@gmail.com (S.H.W.); joe@tigertech.solutions (M.J.W.); dustin@tigertech.solutions (D.H.); evadanielle@gmail.com (E.D.W.); steph.hendricks@gmail.com (S.H.); 3Department of Anesthesiology, Mount Sinai Medical Center, Miami, FL 33140, USA; 4Department of Anesthesiology, Wertheim School of Medicine, Florida International University, Miami, FL 33199, USA; 5Miami Beach Anesthesiology Associates, Miami Beach, FL 33140, USA; 6Department of Physical Therapy, Miller School of Medicine, University of Miami, Miami, FL 33136, USA; j.girardi@med.miami.edu; 7United States Army Research Laboratory, DEVCOM, Adelphi, MD 20783, USA; stephen.j.lee28.civ@mail.mil; 8School of Kinesiology and Recreation, Exercise Science, Illinois State University, Normal, IL 61761, USA

**Keywords:** training load, ANS, American football, collegiate, overtraining, accelerometer, exercise cardiac load, deterioration

## Abstract

Sport coaches increasingly rely on external load metrics for designing effective training programs. However, their accuracy in estimating internal load is inconsistent, and their ability to predict autonomic nervous system (ANS) deterioration is unknown. This study aimed to evaluate the relationships between internal and external training load metrics and ANS recovery and function in college football players. Football athletes were recruited from a D1 college in the southeastern US and prospectively followed for 27 weeks. Internal load was estimated via exercise cardiac load (ECL; average training heartrate (HR) × session duration) and measured with an armband monitor equipped with electrocardiographic capabilities (Warfighter Monitor^TM^ (WFM), Tiger Tech Solutions, Miami, FL, USA). External load was estimated via the summation and rate of acceleration and decelerations as measured by a triaxial accelerometer using the WFM and an accelerometer-based (ACCEL) device (Catapult Player Load, Catapult Sports, Melbourne, Australia) worn on the mid-upper back. Baseline HR, HR variability (HRV) and HR recovery served as the indicators for ANS recovery and function, respectively. For HRV, two, time-domain metrics were measured: the standard deviation of the NN interval (SDNN) and root mean square of the standard deviation of the NN interval (rMSSD). Linear regression models evaluated the associations between ECL, ACCEL, and the indicators of ANS recovery and function acutely (24 h) and cumulatively (one- and two-week). Athletes (n = 71) were male and, on average, 21.3 ± 1.4 years of age. Acute ECL elicited stronger associations for 24 h baseline HR (*R*^2^ 0.19 vs. 0.03), HR recovery (*R*^2^ 0.38 vs. 0.07), SDNN (*R*^2^ 0.19 vs. 0.02) and rMSSD (*R*^2^ 0.19 vs. 0.02) compared to ACCEL. Similar results were found for one-week: 24 h baseline HR (*R*^2^ 0.48 vs. 0.05), HR recovery (*R*^2^ 0.55 vs. 0.05), SDNN (*R*^2^ 0.47 vs. 0.05) and rMSSD (*R*^2^ 0.47 vs. 0.05) and two-week cumulative exposures: 24 h baseline HR (*R*^2^ 0.52 vs. 0.003), HR recovery (*R*^2^ 0.57 vs. 0.05), SDNN (*R*^2^ 0.52 vs. 0.003) and rMSSD (*R*^2^ 0.52 vs. 0.002). Lastly, the ACCEL devices weakly correlated with ECL (rho = 0.47 and 0.43, *p* < 0.005). Our findings demonstrate that ACCEL poorly predicted ANS deterioration and underestimated internal training load. ACCEL devices may “miss” the finite window for preventing ANS deterioration by potentially misestimating training loads acutely and cumulatively.

## 1. Introduction

In athletics, acute and chronic training loads strongly dictate sports performance and the length of adequate recovery [1,2]. Thus, accuracy of estimating training loads with and between sessions is paramount in avoiding overtraining-related injuries and preventing autonomic nervous system (ANS) deterioration [3,4]. Poor recovery of the ANS affects neuromotor function, eliciting negative effects on speed, strength, power, agility, etc. [5,6]. Sport coaches have increasingly relied on various objective and subjective measures to estimate internal and external training loads [7]. Unfortunately, scientific evidence remains inconclusive on which type training load (internal and/or external) and metric best predicts ANS recovery and function [7,8].

Measures of internal load systems provide data on several physiological responses to exercise training like heart rate (HR), heart rate variability (HRV), O_2_ consumption and CO_2_ production [9]. While utilizing gold-standard measures of internal load are ideal (e.g., blood analyses, indirect calorimetry), they are impractical for tracking the internal load daily, especially for larger sports teams like those in American football (≥100 athletes). As such, football coaches rely on devices equipped with electrocardiographic capabilities like HR monitors that provide data on baseline HR, average HR, peak HR and HR recovery. These HR metrics are often used to estimate exercise intensity (e.g., %HR reserve, training impulse (TRIMP), age-predicted maximum HR). However, because these intensity estimates are derived from assumption-based equations and not necessarily applicable athletes [10,11,12], their accuracy in determining internal training load is significantly reduced. Exercise cardiac load (ECL) is a newly validated metric that provides a more accurate estimate of internal load using average training HR and session duration [13]. Recent work shows ECL outperforms commonly used HR-based metrics in estimating the influence of acute and cumulative internal training load on ANS recovery and function [3].

External measures of training estimate total physical work done using performance metrics like power output, running velocity, resistance load, etc. [14]. In theory, external load closely aligns with measures of internal load as scientific research previously established a strong, linear relationship between physical work and the subsequent physiological responses [15]. Specifically, a longer duration and higher intensity of work performed result in larger physiological responses. If sound, this theoretical relationship provides sport coaches with sports performance data that are familiar, practical and more easily translated to training programs [14]. One of the most common external load measures utilized by sport coaches are acceleration-based devices (ACCEL), which track acceleration and deceleration of movements performed by athletes. From these data, training load is estimated using the number and rate of accelerations. These data represent the volume of physical work performed by each player. Recent research has demonstrated, however, significant flaws in using ACCEL to estimate external load as it cannot account for movements occurring at a constant speed [8,16]. As such, these devices may significantly underestimate acute and chronic training loads, thereby increasing the risk of ANS dysfunction.

Therefore, the purpose of this study is two-fold. In a sample of Division I male collegiate football players, we first investigated the influence of acute and cumulative internal (ECL) and external loads (ACCEL) on ANS recovery and function. Specific indicators for ANS recovery and function included baseline HR, HR variability (HRV) and HR recovery, respectively. Second, we evaluated the agreement between ECL and ACCEL. We hypothesized that ECL would outperform ACCEL in predicting ANS recovery and function, yielding stronger associations for both acute and cumulative exposures. Additionally, we posited that ACCEL would elicit a weak association with ECL.

## 2. Materials and Methods

### 2.1. Study Design

This study employed a prospective cohort study in a natural setting among a sample of Division I (D1) collegiate male football players in the southeastern region of the United States. Internal physiological load was measured via an exercise cardiac load metric (ECL), while external load was assessed via an accelerometer-based device. ANS recovery and function were tracked via measuring baseline HR and HR recovery 24 h post training across 25 weeks of pre-season, in-season and out-of-season training.

### 2.2. Subjects

The collegiate football players were recruited from a group of athletes pre-selected by the coaching staff. The selected subjects were “starters” which represented athletes that played nearly every regulation game for most of its duration. Seventy-one subjects voluntarily consented to participate following a fully informed discussion of the study protocols, risks, benefits and right to withdrawal. On average, athletes were 21.3 (±1.4; 18.0–24.0 y) years of age, weighed 100.3 (±23.2; 85.3–143.3 kg) kg, and 58.2% and 36.3% were classified as overweight and obese, respectively.

### 2.3. Exercise Training Sessions

The off-, pre- and in-season training occurred from May to December 2022, lasting a total of 27 weeks (see Figure 1). The 10-week off-season included two 4-week training blocks (Training Camps I and II) with each block separated by a rest. Next, the athletes participated in their 4-week Pre-Season Camp followed by the 13-week in-season training. All athletes followed the same training program, which occurred during football practice sessions (n = 241) and included strength and power resistive exercises, short-distance sprint intervals, aerobic and agility training. The intensity, exercises and duration fluctuated each week; however, on average, the sessions lasted 119.4 (±43.4 87.4–237.3 min) minutes (Table 1).

### 2.4. Internal Load Using WFM

Internal load represented the physiological demand endured throughout each training session and was estimated using a previously established exercise cardiac load (ECL) metric. ECL is the product of the average training HR and duration of training sessions and is expressed in total heartbeats (see equation below):ECL (total heartbeats) = Average HR (bpm) ∙ Session Duration (min)

The average training HR was calculated using only HRs ≥ 85 bpm as this represented an “active training” state. Session duration is the amount of time, in minutes, that the athlete spent above 85 bpm. When HRs fell below 85 bpm, the athletes were classified as “non-active”. Given that the dose-response HR response to exercise intensity and duration parallels other cardiovascular and metabolic responses to exercise like blood pressure, skeletal muscle blood flow and substrate utilization, ECL accurately represents the athletes’ physiological demand of training sessions [9].

### 2.5. HR Measurement

ECL was measured using an armband monitor with electrocardiography and inertial movement capabilities. Throughout each practice session, athletes wore the armband monitors on the posterior aspect of their right upper arm, fastened with an elastic band around the middle of the biceps muscle. The armband monitors (Warfighter Monitor^TM^ (WFM), Tiger Tech Solutions, Inc., Miami, FL, USA) were previously validated in diverse populations including athletes [4,13,17,18] and were worn throughout the entire training session.

### 2.6. External Load using WFM and Catapult

External load represented the physical work performed throughout each training session and was estimated using the summation of instantaneous changes in acceleration measured by a triaxial accelerometer [19,20]. The ACCEL devices used two different equations to estimate external load as presented below:$$\mathrm{WFM}\;\mathrm{External}\;\mathrm{Load}=\sum_{t=0}^{t=n}\sqrt{\frac{{\left({ax}_{t}-{ax}_{t-1}\right)}^{2}+{\left({ay}_{t}-{ay}_{t-1}\right)}^{2}+{\left({az}_{t}-{az}_{t-1}\right)}^{2}}{100}}$$
$$\mathrm{Catapult}\;\mathrm{External}\;\mathrm{Load}=\sum_{t=0}^{t=n}\sqrt{{\left({ax}_{t}-{ax}_{t-1}\right)}^{2}+{\left({ay}_{t}-{ay}_{t-1}\right)}^{2}+{\left({az}_{t}-{az}_{t-1}\right)}^{2}}$$
where ${ax}_{t}$, ${ay}_{t}$ and ${az}_{t}$ are the time-varying acceleration values measured from a triaxial accelerometer. Using a Cartesian coordinate system, ax respresents accelerations in the the *x* direction; similarly, ay and az represent accelerations in the *y* and *z* directions, respectively. The measurements had a total duration of *n* + 1. The subscript *t* represents a single value of acceleration, and *t* − 1 represents the sampled value right before the value at *t*. 

### 2.7. ANS Recovery 24 h Post Training

Next-day baseline HR represented the recovery of the ANS 24 h post training. Baseline HR was measured early in the morning prior to the start of the next day’s training session (0600–0700). Per established protocols, baseline HR was measured following at least 4 min of inactivity and required the athletes to sit nearly motionless for an additional 5 min to collect a “resting” HR [21]. 

#### Heart Rate Variability

Like baseline HR, HRV was measured in the morning (0600–0700), prior to the start of the football training session and 24 h after the start of the previous day’s training session. Per established protocols [21], HRV was obtained following at least 4 min of inactivity. Two HRV time-domain metrics were measured which assessed the changes in the inter-beat interval including RR and NN intervals. RR intervals were the time between R waves on consecutive QRS complexes, and NN intervals were noise-free RR intervals. From these data, the two separate time-domain metrics were derived, including SDNN (standard deviation of the NN interval) and rMSSD (the root mean square of the standard deviation of the NN interval). These HRV time-domain metrics are shown to reflect the parasympathetic autonomic output [22].

### 2.8. ANS Function 24 h Post Training

Next-day HR recovery represented ANS function 24 h post training. HR recovery was defined as the reduction in HR within the first 30 s of rest following a bout of exercise. This time interval was selected as this is when HR exhibits the greatest rate of change. HR recovery was quantified for all resting intervals and then averaged [23].

Importantly, baseline HR and HR recovery were only measured on days preceded by a training session. Thus, on days following one or more rest days, baseline HR and HR recovery were not measured. Including baseline HR, HR recovery and HR values following rest days would likely dilute the association and not accurately represent the ANS recovery and function in response to training.

### 2.9. Statistical Analysis

The current study first evaluated the associations between ECL, ACCEL and indicators of ANS recovery and function, including 24 h baseline HR, HR recovery and SDNN and rMSSD. Second, we evaluated the association between ECL and ACCEL. Three separate analyses were performed for all associations to evaluate the influence of acute (24 h post training) and cumulative (one- and two-week) exposures to training loads estimated via ECL and ACCEL. For acute training loads (e.g., daily sessions), ECL, ACCEL, baseline HR, HR recovery and HRV (SDNN and rMSSD) were averaged across daily sessions for each of the 25 weeks of pre-, in- and off-season training. For the cumulative exposures, the metrics were averaged over one- and two-week training periods. Associations were quantified using linear regression models and were performed separately for each metric. Normality of the conditional distributions was assessed via the Kolmogorov–Smirnov test and was deemed normally distributed. For all models, *β* coefficients and standard errors were estimated, and the a priori threshold for statistical significance for two-sided hypotheses was set at α = 0.05. Statistical analyses performed in MATLAB, version 2021b (MathWorks, Natick, MA, USA).

## 3. Results

Table 1 presents the average internal and external loads in addition to ANS recovery and function across the 27-week season. For internal load, the average ECL reached for athletes during training sessions was 17,110.3 (±5128.6), and the total heart beats and ranged from 11,771.1 to 34,362.9 total heart beats. The ACCEL Catapult device showed an average external load of 328.6 (±113.0), while the WFM ACCEL device showed an average external load of 630.2 (±239.2). For ANS recovery measured 24 h post exercise training, the athletes exhibited, on average, a baseline HR of 62.8 (±6.8) bpm, SDNN of 94.2 ms (±18.3) and rMSSD of 72.9 ms (±7.4).

The linear regression coefficients for the associations between internal and external load on the acute effect of ANS recovery and function are presented in Table 2. Internal load, estimated using ECL, was significantly associated with baseline HR, HR recovery, SDNN and rMSSD 24 h post exercise training. Compared to both ACCEL devices, ECL elicited stronger associations for baseline HR (*R*^2^ 0.19 vs. 0.03), HR recovery (*R*^2^ 0.38 vs. 0.07), SDNN (*R*^2^ 0.19 vs. 0.02) and rMSSD (*R*^2^ 0.19 vs. 0.02). Additionally, the estimates appeared more precise for the internal load metric given the smaller standard errors and narrower 95% confidence intervals. The associations between ACCEL and ANS recovery and function did not reach statistical significance.

Table 3 presents the associations between ECL and ACCEL on the one-week cumulative effect of ANS recovery and function. Both ECL and ACCEL were significantly associated with baseline 24 h HR, HR recovery and HRV following one week of exercise training. Despite reaching statistical significance, the ACCEL devices exhibited weaker associations with ANS recovery and function relative to internal load: 24 h baseline HR (*R*^2^ 0.05 vs. 0.48), HR recovery (*R*^2^ 0.55 vs. 0.05), SDNN (*R*^2^ 0.05 vs. 0.47) and rMSSD (*R*^2^ 0.05 vs. 0.47). While the *β* coefficients for the ACCEL were higher compared to the internal load, the stability and precision of the linear regression model appear low given the larger standard errors and wider 95% confidence intervals.

The associations between ECL and ACCEL on the two-week cumulative effect of ANS recovery and function are compared in Table 4. ECL showed statistically significant and stronger associations with baseline 24 h HR, HR recovery and HRV following two weeks of exercise training relative to the ACCEL devices: baseline HR (*R*^2^ 0.52 vs. 0.003), HR recovery (*R*^2^ 0.57 vs. 0.05), SDNN (*R*^2^ 0.52 vs. 0.003) and rMSSD (*R*^2^ 0.52 vs. 0.003 vs. 0.002). The weaker associations reported for ACCEL did not reach statistical significance. Additionally, the association with acute and one-week accumulative effects appeared more precise given the smaller standard errors and narrower 95% confidence intervals. Lastly, the Pearson correlation coefficients representing the association between ECL and ACCEL showed a positive, statistically significant relationships for both the WFM and Catapult devices (ρ = 0.43 and 0.47, *p* < 0.0005, respectively).

## 4. Discussion

The study evaluated whether an internal (ECL) or external (ACCEL) training load metric would best predict ANS deterioration in a sample of male Division I football athletes. Specifically, we investigated the relationship between internal (ECL) and external (ACCEL) loads on ANS recovery and function across acute (24 h) and cumulative (one- and two-week) periods. Following, we evaluated the relationship between the ECL and ACCEL metrics. The major findings of this study were: (1) ACCEL demonstrated poor predictive power with all indicators of ANS recovery and function for both acute and cumulative exposures; (2) unlike ACCEL, ECL showed strong predictive power for all indicators of ANS recovery and function that strengthened over time; and (3) ACCEL exhibited a weak correlation with ECL and appeared to underestimate the exercise training load prescribed during routine football practice sessions. The differential relationships with ANS recovery, function and ECL observed in the current study may reflect significant limitations of ACCEL utilization in estimating training load impact. Our findings suggest that tracking the internal load may preferentially enhance sports performance and prevent overtraining in collegiate American football players. 

### 4.1. Internal Load vs. External Load on ANS Deterioration

An important and novel aspect of the current study was the large differences in the abilities of ACCEL and ECL in predicting ANS deterioration. ACCEL showed weak, mostly non-significant relationships with the selected indicators of ANS recovery and function, including baseline HR, HR recovery, SDNN and rMSSD. Strengths of the associations ranged from an R^2^ of 0.002 to 0.05, and the linear models exhibited large standard errors and wide 95% confidence intervals. The latter observation indicates high imprecision of the ACCEL metric in predicting ANS deterioration. Additionally, these relationships either decreased or remained constant across the 24 h, one-week and two-week acute and cumulative exposures, respectively. Unlike ACCEL, ECL demonstrated strong and statistically significant relationships with ANS recovery and function. Athletes exposed to high ECLs showed elevated baseline HR, depressed HRV and slower recovery during 30 s intervals. Importantly, these relationships strengthened for all outcomes (R^2^ 0.19 to 0.52) across both acute and cumulative exposures. The relationships between ECL and ANS deterioration observed in the current study closely align with our previous work among collegiate football athletes that also showed the ECL metric exhibited strong predictive power of ANS deterioration for acute and cumulative exposures [3,13]. There are two significant concerns highlighted by these observations, specifically the poor predictive power of ACCEL. First, if undetected, ANS deterioration likely leads to non-functional overreaching and, inevitably, overtraining syndrome [6]. Once overtrained, depending on the severity, athletes require weeks to months of rest and become predisposed to repeatedly overtrain following recovery. Overtraining is rather prevalent among elite-level athletes, with former studies reporting nearly 60% of elite athletes experience signs, symptoms and/or diagnosis of overtraining at least once throughout their sports career [24]. Strong evidence indicates that overtraining is consequent to athletes performing a combination of high-intensity and/or high-volume exercise training with insufficient recovery [6,24,25,26]. This emphasizes the necessity of coaches to utilize devices that accurately estimate training load, either internal or external. A related concern is the ever-growing reliance on ACCEL devices by sport coaches. Coaches utilize ACCEL devices to estimate training load and track its effect on athlete performance and recovery [14,16]. However, the weak ability of ACCEL devices to predict ANS deterioration may suggest significant error in ACCEL devices in accurately estimating training load.

### 4.2. Internal Load vs. External Load on Training Volume

Another unique aspect of the current study was the evaluation of the relationship between ACCEL and ECL. The ACCEL metric, for both devices, exhibited significant, positive correlations, of moderate strength, with ECL. Moreover, it appeared that ACCEL underestimated the exercise training load to which the athletes were exposed. Intuitively, the ACCEL and ECL metrics are correlated, as they each estimate training load by measuring training volume (e.g., physical work performed), albeit differently. However, theoretically, ECL and ACCEL should exhibit a strong relationship if they are, in fact, both measuring the same thing, training load [15,27]. Here, our observation demonstrates that ACCEL is not an accurate surrogate for estimating training load via physiological responses. ECL measured the internal load via the total number of beats accumulated during a training session such that a higher ECL equated to a larger training volume achieved by a combination of increased exercise intensity and/or duration. ACCEL measured the external training load via the total number and rate of accelerations performed during a training session such that more and faster accelerations achieved equated to a larger training volume. The moderate associations between ACCEL and ECL in addition to ANS recovery and function potentially suggest significant measurement flaws of ACCEL devices in estimating exercise training load. Because the ACCEL device only measures changes in acceleration, any physical work performed at a constant speed, either aerobic or anaerobic, remains unaccounted for, ultimately underestimating a training load to which an athlete is exposed [8]. This flaw is especially problematic for football athletes in positions that include running routes performed with minimal changes in acceleration like those of wide receivers and cornerbacks [8]. The underestimated training load, unbeknownst to coaches, likely results in coaches incorporating more higher-intensity and/or higher-volume exercises during practices. This misestimation in training load is critical as ANS deterioration may occur within 24 h and, if left unnoticed, worsens over time with eventual decrements in sport performance [1,6,28]. In support, our most recent study showed that cumulative exposures to misestimated training loads, estimated via ECL, coupled with insufficient recovery led to deterioration of maximum running speed among “heavy running” positions like wide receivers and cornerbacks throughout a 27-week collegiate football season^3^. That is, the maximum speed at which an athlete achieved during training sessions decreased during pre-, in- and off-season. 

### 4.3. Practical Implications

These findings highlight the importance of utilizing metrics that accurately measure the physiological stresses athletes endure during training sessions, especially at the elite level given the high frequency of training. Moreover, our study may suggest that tracking the internal training load, rather than or in addition to the external training load, is the best method for enhancing sport performance and avoiding overtraining. Internal load metrics like ECL are non-specific regarding movement patterns. ECL merely collects data on the response of the cardiovascular system, specifically the total number of contractions of the cardiac muscle. This provides coaches a reliable metric in that the cardiovascular system will continuously respond in proportion to physical work performed given it is responsible, in part, for increasing blood flow and O_2_ to the working skeletal muscles [29,30]. Moreover, when using devices with electrocardiographic capabilities like the WFM, coaches can track the athlete’s physiological tolerance and recovery to daily and weekly exercise sessions. Subsequently, the data can be used to individualize an athlete’s training program to their physiological tolerance profile potentially enhancing sport performance and preventing overtraining. Further, using an internal load metric like ECL may provide indicators of other external stressors influencing ANS recovery and function like insufficient sleep [31], emotional turmoil [32,33], prolonged high cognitive load [34], etc. The negative effect of these stressors on the ANS is well established. While the ECL metric cannot identify the specific external stressor, it can provide coaches with other avenues for explaining decrements in sports performance and ANS deterioration, especially if the training load is tolerable [35]. We acknowledge that external load metrics like ACCEL may provide useful performance data like velocity, angular momentum, etc., and do not discourage their use, but rather recommend selectively utilizing these devices. For example, ACCEL devices may be more accurate in estimating training load among athletes playing positions that frequently require short, explosive and powerful movement patterns like linemen and rugby players, as documented in previous studies [15]. It is imperative for coaches to understand the measurement limitations when opting for ACCEL devices. Even if ACCEL devices, in some instances, accurately measure training load, they likely cannot directly track an athlete’s physiological response or tolerance to training loads nor preemptively avoid overtraining. As such, ACCEL devices should be used in combination with a non-invasive measure on internal load.

### 4.4. Strengths and Limitations

This study has strengths worth mentioning. First, this study was conducted in a natural setting and, as such, our observations likely reflect the exercise training loads to which football athletes are exposed during practice sessions. Second, the ECL was previously validated for estimating exercise training load and tracking ANS recovery and function in collegiate football athletes, strengthening the confidence in our findings. Third, the WFM and ACCEL devices were worn simultaneously during all practice sessions, ensuring identical training load exposures and allowing for a rigorous comparative analysis of their performance. A limitation of this study was the inclusion of only one Division I collegiate football team in a single geographical location, limiting our generalizability and precluding similar conclusions to be drawn for (1) other sports, (2) professional athletes and (3) female athletes.

## 5. Conclusions

In conclusion, our study demonstrated a few important concepts and identified critical areas of future research. First, ACCEL devices used to externally estimate the training load poorly correlated with a validated measure of the internal training load. Specifically, ACCEL underestimated the exercise training load compared to the ECL. Second, ACCEL poorly predicted ANS deterioration to acute and cumulative exposures to high training loads. These observations suggest that ACCEL devices may “miss” the finite window for preventing ANS deterioration, non-functional overreaching and overtraining. The high prevalence of overtraining signs and symptoms among elite-level athletes emphasizes the need for coaches to utilize devices that accurately estimate the training load, such as the ECL, either as a standalone metric or in addition to external load metrics. We strongly recommend that future studies investigate the performance of both internal and external load metrics across multiple sports and individual positions to better inform coaches on the “best” device(s) for enhancing sports performance and preventing ANS deterioration.

## Figures and Tables

**Figure 1 jfmk-09-00005-f001:**
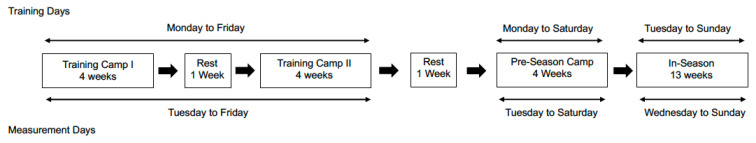
Collegiate American football training schematic.

**Table 1 jfmk-09-00005-t001:** Descriptive data on football training sessions, load metrics and ANS recovery and function.

	Mean (SD)	Median (Min, Max)
Training Sessions		
No. of Practice Sessions	241	-----
Duration of Practice Sessions (min)	119.4 (43.4)	122 (87.4, 237.3)
Load Metrics		
Catapult External Load (a.u.)	328.58 (113.0)	321.5 (34.99, 800.6)
WFM External Load (a.u.)	630.2 (239.2)	610.7 (51.9, 1682.7)
WFM Internal Load (ECL) *	17,110.4 (5128.6)	17,110.3 (11,771.1, 34,362.9)
ANS Recovery and Function		
Baseline (bpm)	62.8 (6.8)	62.3 (44.8, 90.5)
SDNN (ms)	94.2 (18.3)	93.8 (47.3, 189,9)
rMSSD (ms)	72.9 (7.4)	73.1 (23.8, 108.4)
HR Recovery (bpm)	21.3 (5.3)	21.4 (8.4, 46.2)

* ECL is expressed as the total number of heart beats; a.u. = arbitrary units; WFM = Warfighter Monitor; bpm = beats per minute; ms = milliseconds; SDNN = standard deviation of all normal NN intervals; rMSSD = root mean square of successive differences between normal heartbeats; min = minutes.

**Table 2 jfmk-09-00005-t002:** Linear regression coefficients for the separate associations between external and internal load and 24-h acute effect on ANS recovery and function.

	External Load Catapult	*p*-Value	External Load WFM	*p*-Value	Internal Load WFM	*p*-Value
Baseline HR						
*R*^2^	0.03	0.16	0.03	0.17	0.19	<0.0001
*β* coefficient (SE)	0.46 (0.30)		0.88 (0.64)		0.14 (0.01)	
95% CI	(−0.16, 1.02)		(−0.39, 2.12)		(0.11, 0.17)	
HR Recovery						
*R*^2^	0.07	0.0003	0.07	0.0003	0.38	<0.0001
*β* coefficient (SE)	−1.38 (0.38)		−2.94 (0.82)		0.37 (0.02)	
95% CI	(−2.13, −0.62)		(−4.54, −1.35)		(0.34, 0.40)	
HRV—SDNN						
*R*^2^	0.02	0.21	0.02	0.23	0.19	<0.0001
*β* coefficient (SE)	−0.19 (0.15)		−0.39 (0.32)		−0.07 (0.01)	
95% CI	(−0.48, 0.10)		(−1.01, 0.24)		(−0.08, −0.05)	
HRV—rMSSD						
*R*^2^	0.02	0.21	0.02	0.23	0.19	<0.0001
*β* coefficient (SE)	−0.35 (0.28)		−0.71 (0.59)		−0.12 (0.01)	
95% CI	(−0.90, 0.20)		(−1.87, 0.45)		(−0.15, −0.10)	

WFM = Warfighter Monitor; internal load measured via ECL; HRV = heart rate variability; SDNN = standard deviation of all normal NN intervals; rMSSD = root mean square of successive differences between normal heartbeats; *β* = beta SE = standard error; CI = confidence interval.

**Table 3 jfmk-09-00005-t003:** Linear regression coefficients for the separate associations between external and internal load and one-week cumulative effect on ANS recovery and function.

	External Load Catapult	*p*-Value	External Load WFM	*p*-Value	Internal Load WFM	*p*-Value
Baseline HR						
*R*^2^	0.05	0.01	0.05	0.01	0.48	<0.0001
*β* coefficient (SE)	0.51 (0.18)		1.04 (0.37)		0.37 (0.01)	
95% CI	(0.15, 0.87)		(0.30, 1.77)		(0.35, 0.40)	
HR Recovery						
*R*^2^	0.05	0.002	0.05	0.002	0.55	<0.0001
*β* coefficient (SE)	0.74 (0.24)		1.49 (0.48)		0.58 (0.02)	
95% CI	(0.28, 1.2)		(0.54, 2.43)		(0.55, 0.61)	
HRV—SDNN						
*R*^2^	0.05	0.01	0.05	0.006	0.47	<0.0001
*β* coefficient (SE)	−0.25 (0.09)		−0.51 (0.18)		−0.18 (0.01)	
95% CI	(−0.43, −0.08)		(−0.87, −0.15)		(−0.19, −0.17)	
HRV—rMSSD						
*R*^2^	0.05	0.01	0.05	0.006	0.47	<0.0001
*β* coefficient (SE)	−0.47 (0.17)		−0.95 (0.34)		−0.33 (0.01)	
95% CI	(−0.80, −0.14)		(−1.63, −0.28)		(−0.36, −0.31)	

WFM = Warfighter Monitor; internal load measured via ECL; HRV = heart rate variability; SDNN = standard deviation of all normal NN intervals; rMSSD = root mean square of successive differences between normal heartbeats; *β* = beta SE = standard error; CI = confidence interval.

**Table 4 jfmk-09-00005-t004:** Linear regression coefficients for the separate associations between external and internal load and two-week cumulative effect on ANS recovery and function.

	External Load Catapult	*p*-Value	External Load WFM	*p*-Value	Internal Load WFM	*p*-Value
Baseline HR						
*R*^2^	0.003	0.85	0.003	0.85	0.52	<0.0001
*β* coefficient (SE)	0.03 (0.18)		0.07 (0.36)		0.40 (0.01)	
95% CI	(−0.32, 0.39)		(−0.64, 0.78)		(0.38, 0.42)	
HR Recovery						
*R*^2^	0.05	0.001	0.05	0.001	0.57	<0.0001
*β* coefficient (SE)	0.76 (0.23)		1.56 (0.47)		0.61 (0.02)	
95% CI	(0.30, 1.21)		(0.64, 2.5)		(0.58, 0.64)	
HRV—SDNN						
*R*^2^	0.003	0.88	0.003	0.88	0.47	<0.0001
*β* coefficient (SE)	−0.01 (0.09)		−0.03 (0.18)		−0.18 (0.01)	
95% CI	(−0.19, 0.16)		(−0.38, 0.32)		(−0.19, −0.17)	
HRV—rMSSD						
*R*^2^	0.003	0.88	0.002	0.88	0.47	<0.0001
*β* coefficient (SE)	−0.03 (0.16)		−0.05 (0.33)		−0.33 (0.01)	
95% CI	(−0.35, 0.30)		(−0.70, 0.60)		(−0.36, −0.31)	

WFM = Warfighter Monitor; internal load measured via ECL; HRV = heart rate variability; SDNN = standard deviation of all normal NN intervals; rMSSD = root mean square of successive differences between normal heartbeats; *β* = beta SE = standard error; CI = confidence interval.

## Data Availability

Data presented in the current paper are available upon request.

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
