# Peer review of "Internal or External Training Load Metrics: Which Is Best for Tracking Autonomic Nervous System Recovery and Function in Collegiate American Football?"

_jfmk, 2023, doi:10.3390/jfmk9010005_

Round 1

Reviewer 1 Report

Comments and Suggestions for Authors

Dear all,

The manuscript fits with the journal's aim, and the subject reveals good content for researchers and professionals in the field. The reviewed article analyses the concept of being a fascinating topic for competitive sports and high performance that depends on internal and external training loads. However, some minor points are listed below:

Title

The title is long, and it should be concise, specific, and relevant. I suggest the below one:

Internal or external training load metrics: which is best for tracking autonomic nervous system recovery and function for collegiate football players.

Abstract

The abstract provides an accurate description of the components of the work. Nonetheless, there are many abbreviations. Please reduce them as much as possible. In addition, some of these abbreviations were mentioned without mentioning the full name of the abbreviated term:

Line 28: exercise cardiac load (ECL).

Line 35: acceleration-based devices (ACCEL))

Introduction

The introduction has a good and plausible connection to the study methodology and purpose, and it is acceptable and sufficient.

Materials and Methods

The methodological design is appropriate for the research objectives. A well-designed research method provides support for both the technique and the results that are presented.

It will be useful to present the participant characteristics, adding to the age, height, and body mass next details (training experience, players position, how many 'centre backs, midfielders, strikers, wingers, and full backs).

In line 132; ‘The average training HR was calculated using only HRs exceeding ≥ 85 bpm’. Unfortunately, you didn’t mention any about the HRs ≥ 85, or the data-driven intensity thresholds HR ≥ 140, 150, or 160 bpm

Between lines 124-125; the inserted figure doesn't have a sufficiently high resolution and needs a short explanatory title and caption.

In line 129; the authors defined the ECL and present how it was calculated using the equation in line 131. However, in line 149, authors showed that the external load was calculated using two different equations. Could you explain why there are 2 equations for external load and not like Internal load?

In lines 151-152; the equations to estimate external load (𝑾𝑭𝑴 𝑬𝒙𝒕𝒆𝒓𝒏𝒂𝒍 𝑳𝒐𝒂𝒅 and 𝑪𝒂𝒕𝒂𝒑𝒖𝒍𝒕 𝑬𝒙𝒕𝒆𝒓𝒏𝒂𝒍 𝑳𝒐𝒂𝒅) need to be explained (axt, ayt, azt…)

The measured variables and markers are incomplete. I wish if include (total distance, Accelerations, Decelerations, number of accelerations above 2.5 m.s2, volume of high-speed running per minute).

The "content" of the training, as well as the internal stress, need to be presented. Were all the participants instructed to participate in the equivalent identical training loads and exercises?

Results

The results are interesting and give a deep response to the research questions.

All tables require clarification of definitions, notes, and sources utilized within the table. Any acronyms or abbreviations used in the table should be mentioned alphabetically in the notes under the table.

I understand that training is done in groups as much as feasible and individually when necessary; consequently, it is my recommendation that the correlations be performed by positions (center backs, midfielders, strikers, wingers, and full backs), taking into consideration the particulars of the training programme for each.

It was necessary to use modified linear regression coefficients to represent the connections between the duration spent at various intensity thresholds and the subsequent-day autonomic nervous system (ANS) recovery.

Discussion

In the discussion section, please start with the aim of your review study, the strengths of this study, and the chief results and conclusions you found that most of the publications evaluated.

Conclusions

Adequate

References

Need to be checked. I found journal names are written in full name (e.g. references 11, 12 , and 13) and some in abbreviation (e.g. references 5, 6, and 7).  Please use the ACS style guide to be compatible with J Funct Morphol Kinesiol guidelines.

With my best regards, 

Reviewer 2 Report

Comments and Suggestions for Authors

This is a comprehensive study that with a duration of 27 weeks, covers a substantial period of time. It is noteworthy that the reviewer previously had the opportunity to evaluate another study conducted by the same group. Consequently, one may question why a team of 11 authors is unable to produce a manuscript devoid of grammatical errors. Presented below are both significant and minor comments that, as previously advised, should be attentively rectified.

Major

HRV should be regarded not as a standalone measure but as an indication of the slight variations in the time intervals between heartbeats. These fluctuations can be evaluated using various HRV metrics. It is essential to use precise terminology when referring to HRV.

L. 160: you previously highlighted that HRV measures are sensitive to changes in vagal activity. It is advisable to analyse, present, and discuss these measures to provide a comprehensive understanding of the findings.

In Reference 8, the importance of HRV measures is acknowledged even in American football players. However, it is crucial to clarify the specific motivation for conducting the current study.

Discussion section: Reading that section with its roughly 130 lines is demanding. Please, enter subtitles.

Kindly alleviate my concerns regarding the involvement of seven out of eleven authors with WFM. Providing additional context or clarification about their roles and potential impacts on the study would be beneficial.

Minor

Football: Please add: American …

l. 27: How many participants?

l. 29: HR is somewhat misleading here. Could be understood as hours. Please, introduce heart rate (HR)

l. 33: The entire HRV was likely not assessed. Thus, please specify: RMSSD, LF/HF, Tri-Index or similar?

L 35: ACCEL meaning what?

l.35: One can’t correlate something with HRV but with certain measures of HRV. Tell us.

l. 36: likely males?

l. 65: TRIMP?

l. 111: mean ± SD?; range?

l. 123: order?

l. 132: delete: exceeding

l. 133: verb?

l. 143 … was?

l. 144: three references will do. Maybe the youngest and from different groups.

l. 152: If providing equations, the terms need to be explained.

l. 155: … represents recovery of the HR. Not necessarily of the ANS. You don’t know whether sympathetic activity had decreased or vagal activity had increased.

l. 170: Statistics: Please explain, why R² - the coefficient of determination (not correlation) is used.

l. 171 f: One cannot correlate ANS function and HRV.

Maybe, add one-sided / two-sided according to any hypothesis.

l.195: within the 27 weeks?

l. 199: not with HRV but with RMSSD and SDNN?

In Tab. 1 you present SDNN (ms) 94.2 (18.3) and rMSSD (ms) 72.9 (7.4) – values that are different from norm values provided by Nunan et al, DOI: 10.1111/j.1540-8159.2010.02841.x. See also: https://welltory.com/rmssd-and-other-hrv-measurements/  Do you later discuss these differences?

l. 236: Why not keep the order of ECL and then ACCEL?

l. 244: … aspect …. was…

l. 271: Reference(s)?

l. 275: what is the difference with l. 244?

l. 282: Pls, suggest types of sport better suited for ACCEL measurements. Boxing? Volleyball?

l. 345: … and one location.

Comments on the Quality of English Language

Minor typos, minor grammar. See comments.

Round 2

Reviewer 2 Report

Comments and Suggestions for Authors

Thanks for revising the ms.

Please, address two further comments:

Think about whether one can measure a HRV index. You can measure HR and body temperature, but not an index.

Please, unify the order of rMSSD and SDNN, throughout. And decide for rMSSD or RMSSD.
